# IRF3 Activation in Mast Cells Promotes FcεRI-Mediated Allergic Inflammation

**DOI:** 10.3390/cells12111493

**Published:** 2023-05-28

**Authors:** Young-Ae Choi, Hima Dhakal, Soyoung Lee, Namkyung Kim, Byungheon Lee, Taeg Kyu Kwon, Dongwoo Khang, Sang-Hyun Kim

**Affiliations:** 1Cell & Matrix Research Institute, Department of Pharmacology, School of Medicine, Kyungpook National University, Daegu 41944, Republic of Korea; wizchoi@knu.ac.kr (Y.-A.C.); dhakalhima@gmail.com (H.D.); nortonnklab@gmail.com (N.K.); 2Immunoregulatory Materials Research Center, Korea Research Institute of Bioscience and Biotechnology, Jeongeup 56212, Republic of Korea; sylee@kribb.re.kr; 3Department of Biochemistry and Cell Biology, School of Medicine, Kyungpook National University, Daegu 41944, Republic of Korea; leebh@knu.ac.kr; 4Department of Immunology, School of Medicine, Keimyung University, Daegu 42601, Republic of Korea; kwontk@dsmc.or.kr; 5Department of Physiology, School of Medicine, Gachon University, Incheon 21999, Republic of Korea

**Keywords:** allergic inflammation, histamine, histidine decarboxylase, interferon regulatory factor 3, mast cells

## Abstract

(1) Background: This study aims to elucidate a novel non-transcriptional action of IRF3 in addition to its role as a transcription factor in mast cell activation and associated allergic inflammation; (2) Methods: For in vitro experiments, mouse bone-marrow-derived mast cells (mBMMCs) and a rat basophilic leukemia cell line (RBL-2H3) were used for investigating the underlying mechanism of IRF3 in mast-cell-mediated allergic inflammation. For in vivo experiments, wild-type and *Irf3* knockout mice were used for evaluating IgE-mediated local and systemic anaphylaxis; (3) Results: Passive cutaneous anaphylaxis (PCA)-induced tissues showed highly increased IRF3 activity. In addition, the activation of IRF3 was observed in DNP-HSA-treated mast cells. Phosphorylated IRF3 by DNP-HSA was spatially co-localized with tryptase according to the mast cell activation process, and FcεRI-mediated signaling pathways directly regulated that activity. The alteration of IRF3 affected the production of granule contents in the mast cells and the anaphylaxis responses, including PCA- and ovalbumin-induced active systemic anaphylaxis. Furthermore, IRF3 influenced the post-translational processing of histidine decarboxylase (HDC), which is required for granule maturation; and (4) Conclusion: Through this study, we demonstrated the novel function of IRF3 as an important factor inducing mast cell activation and as an upstream molecule for HDC activity.

## 1. Introduction

Allergic diseases affecting approximately 25% of people of all ages are becoming a global concern due to their increasing prevalence in recent decades [1]. Representative allergic diseases include asthma, allergic rhinitis, food allergy, and atopic dermatitis [2,3]. Mast cells are the primary cell type contributing to allergic diseases and, after activation, secrete various allergic inflammatory mediators that lead to inflammation [4]. Furthermore, activating specific signaling pathways further enhances the secretion of these allergic mediators. Thus, it is necessary to elucidate the detailed molecular mechanism of allergic inflammation to develop strategies for effectively preventing and treating mast-cell-associated allergic diseases.

Mast cell signaling is initiated by multivalent antigens interacting with the specific IgE bound to the receptor FcεRI [5]. This results in the phosphorylation of the immunoreceptor tyrosine-based activation motifs of the β and γ subunits of the adjacent FcεRI receptors, and sequentially, the Src family kinases Lyn and Fyn activate Syk. Phosphorylated Syk activates the downstream signaling proteins, such as phosphoinositide 3-kinase (PI3K) and phospholipase Cγ (PLCγ), triggering intracellular calcium release [6]. This signaling activates transcription factors and induces allergic mediators’ secretion, including histamine, proteases, cytokines, and arachidonic acid metabolites stored in cytoplasmic granules [4]. Histamine, a primary mediator of IgE-FcεRI-mediated allergic inflammation, is synthesized via histidine decarboxylation through the enzymatic activity of histidine decarboxylase (HDC) [4,7,8]. Studies have shown that Hdc-deficient mice fail to synthesize histamine and produce IgE-mediated allergic reactions [4,8]. Therefore, modulating the secretion of allergic inflammatory mediators, including the activity of HDC, represents a strategy to control mast-cell-mediated allergic inflammation [7].

Growing evidence suggests the role of transcription factors in regulating mast-cell-mediated allergic inflammation [8,9]. Interferon regulatory factors (IRFs) are a family of transcription factors comprising nine members (IRF1–IRF9) that reside in the cytoplasm of various types of cells [10]. IRFs regulate the innate and adaptive immune response as well as multiple interferon-induced signaling pathways [10]. Among them, IRF3 contributes to the regulation of the adaptive immune response and is associated with inflammatory conditions and allergic diseases [11,12]. The phenotypic alteration of IRF3 impaired the house-dust-mite-induced airway allergy by regulating T helper 2 (Th2) responses and dendritic cell maturation [13,14]. In addition, *Irf*-deficient mice were unable to produce IgE during ovalbumin (OVA)-induced adaptive immune response, which is essential for generating mast-cell-mediated allergic inflammation [15]. Despite these studies, the role and molecular mechanism of IRF3 in mast-cell-associated allergic inflammation are still unclear. In the present study, for the first time, we demonstrate the importance of IRF3 activity in controlling mast-cell-mediated allergic inflammation and how it regulates mast cell function.

## 2. Materials and Methods

### 2.1. Ethics Statement

Animal care and treatment were performed as per the guidelines of the Public Health Service Policy on the Humane Care and Use of Laboratory Animals. In addition, animal experiments were approved by the Institutional Animal Care and Use Committee of Kyungpook National University with an approved IRB protocol (IR2020-0001-123).

### 2.2. Mouse Breeding and Maintenance

B6;129S6-Bcl2L12/*Irf3*<tm1Ttg>/TtgRbrc, C57BL/6J (*Irf3* KO) mice were obtained from the RIKEN BioResource Research Center (Kyoto, Japan). The C57BL/6J strain *Irf3* knockout (KO) mice were bred under specific pathogen-free conditions at Kyungpook National University. These *Irf3* KO mice were back-crossed for 20 generations onto the BALB/c (Dae-Han, Daejeon, Republic of Korea) background to generate homozygous animals that were free of background effects on the phenotypes. Further experiments were performed using wild-type (WT) and *Irf3* KO mice. Mice were maintained in a laminar airflow room maintained at 22 °C ± 2 °C, with relative humidity of 55% ± 5%, and 12 h light:dark cycles with food and water ad libitum.

### 2.3. Reagents

Dinitrophenyl-human serum albumin (DNP-HSA), anti-DNP-IgE, *o*-phthaldialdehyde, and 4-nitrophenyl N-acetyl-β-d-glucosamide were purchased from Sigma-Aldrich (St. Louis, MO, USA). The protease inhibitor cocktail and phosphatase inhibitor cocktail were purchased from Roche (Mannheim, Germany). IL-3 and stem cell factor (SCF) were purchased from PeproTech (EC, London, UK). The signaling inhibitors, PP2 and Akt inhibitor IV were purchased from Sigma-Aldrich, and U 73122 and LY 294002 (ab120243) were from Abcam (Cambridge, UK). The antibodies for anti-IRF3 and p-S396-IRF3 were obtained from Cell Signaling (Beverly, MA, USA), lamin B1 was obtained from Santa-Cruz Biotechnology (Dallas, TX, USA), and anti-HDC was purchased from Invitrogen (Waltham, MA, USA).

### 2.4. Cell Isolation, Culture, and Maintenance

To isolate mouse bone-marrow-derived mast cells (mBMMCs), the femurs and tibias of BALB/c mice were flushed with RPMI-1640 medium. Then, the collected cells were cultured in RPMI-1640 medium supplemented with 10% heat-inactivated FBS, 1× antibiotic-antimycotic, 1 mM sodium pyruvate, 4 mM L-glutamine, 25 mM HEPES, MEM-NEA, 50 µM 2-ME, 10 ng/mL IL-3, and 2 ng/mL SCF for 4 weeks. mBMMC maturation was determined via flow cytometry. More than 80% of FcεRI^+^c-kit^+^ cells were used for further experiments [16]. RBL-2H3 cells (basophilic leukemia cell; American Type Culture Collection CRL-2256) were cultured at 5% CO_2_ and 37 °C in DMEM supplemented with 10% heat-inactivated FBS and 1× antibiotic–antimycotic.

### 2.5. IRF3 Knockdown and Overexpression in mBMMCs and RBL-2H3

*Irf3* siRNA (mouse, s79433 and rat, s146864), *Bcl2l12* siRNA (mouse, s231935), and control siRNA were purchased from Ambion (Austin, TX, USA). The *Irf3* and control siRNAs were introduced into mBMMCs (40 pmol/3 × 10^6^ cells/well in a 6-well plate) and RBL-2H3 (40 pmol/2 × 10^6^ cells/well in a 6-well plate) using the Neon Electroporation System (Invitrogen) following the manufacturer’s protocols. Cells were transfected in an antibiotic-free growth medium for 24 h. Likewise, mBMMCs were also transfected with *Bcl2l12* siRNA. After incubation, the transfected cells were used for the following experiments.

mBMMCs and RBL-2H3 cells were transfected with pCS107 (vector) and pCS107-IRF3 plasmid (wt*Irf3*) DNA. The accession number for the mouse is NM_016849.4, and rat is NM_001006969.1. mBMMCs (0.8 μg/3 × 10^6^ cells/well in a 6-well plate) for 36 h, and RBL-2H3 (1 μg/1 × 10^6^ cells/well in a 6-well plate) were transfected for 24 h, respectively, using the above transfection protocol. After transfection, the cells were used for further experiments.

### 2.6. Induction and Assessment of Passive Cutaneous Anaphylaxis (PCA)

WT and *Irf3* KO mice (*n* = 5/group) were sensitized with an intradermal injection of 0.5 µg/site anti-DNP-IgE into the ear to induce a PCA reaction and then monitored for 48 h. After that, there were challenged with the mixed solution of DNP-HSA (1 mg/mouse) and 4% Evans blue (1:1) via intravenous injection into the tail vein. After 30 min, the mice were euthanized, and ear thickness was measured. Both ears were collected and digested in 1 mL of 1 M KOH and 4 mL of an acetone and phosphoric acid (5:13) mixture. The amount of dye leaked by anaphylaxis was determined colorimetrically using spectrophotometry at 620 nm. Anti-DNP-IgE was used as an antigen-specific IgE source, DNP-HSA as an antigen (Ag), and Evans blue for vasopermeability detection.

### 2.7. Induction and Assessment of OVA-Induced Active Systemic Anaphylaxis (ASA)

WT and *Irf3* KO mice (*n* = 5/group) were injected intraperitoneally with a solution of OVA (100 μg) and alum adjuvant (2 mg) or PBS on days 0 and 7, as previously described, to establish an ASA mouse model [16]. On day 14, the mice were challenged with an intraperitoneal injection of OVA (200 μg). Rectal temperatures were recorded at 10 min intervals for 90 min by inserting a thermometer probe (Testo, Titisee, Germany, cat# 925). After 90 min, mice were sacrificed, and serum was separated from the blood to analyze IgE, OVA-specific IgE, histamine, and IL-4 levels.

### 2.8. Toluidine Blue Staining and Mast Cell Number Counting

Formalin-fixed mouse ear tissue was made into paraffin blocks and then sectioned at 5 μm thickness. The tissue section was stained with Toluidine blue dye. Mast cells were counted in five random sites for each tissue section. Images were acquired at ×200 magnification under Carl Zeiss microscope (Jena, Germany).

### 2.9. Immunofluorescence Staining

mBMMCs (5 × 10^5^ cells/well) were seeded into Nunc Lab-Tek eight-well chamber slides and sensitized with anti-DNP-IgE (50 ng/mL) overnight and then stimulated with DNP-HSA (100 ng/mL) for the indicated time. Cells were fixed, permeabilized, and incubated with phospho-S^396^IRF3 and tryptase antibodies. Then, the cells were incubated with Alexa 488 or 594-labeled anti-rabbit or mouse secondary antibody (Invitrogen). Images were acquired at ×200 magnification using fluorescence microscopy (Carl Zeiss).

### 2.10. Western Blotting

mBMMCs and RBL-2H3 cells were sensitized with anti-DNP-IgE (50 ng/mL) overnight and then stimulated with DNP-HSA (100 ng/mL) for 30 min with or without 1 h pre-treatment with various signaling inhibitors. Total cell lysate or nuclear protein from the cells was collected and subsequently subjected to Western blotting as described in the previous report [17]. The information for the used antibodies is listed in Appendix A. Immunodetection was performed using an enhanced chemiluminescence detection kit (Amersham, Piscataway, NJ, USA). Images were detected using the Syngene G:Box Chemi XRQ gel doc system (Bangalore, India).

### 2.11. Enzyme-Linked Immunosorbent Assay (ELISA)

mBMMCs and RBL-2H3 cells were sensitized with anti-DNP-IgE (50 ng/mL) overnight. Then, mBMMCs were stimulated with DNP-HSA (100 ng/mL) for 6 h, and RBL-2H3 cells were stimulated with DNP-HSA (100 ng/mL) for 8 h. Each assay was performed using an ELISA kit (BD Biosciences, Franklin Lakes, NJ, USA) according to the manufacturer’s protocol. Used kits are listed in Appendix A.

### 2.12. Mast Cell Degranulation Assays

mBMMCs and RBL-2H3 cells were sensitized with anti-DNP-IgE (50 ng/mL) overnight. Then, mBMMCs were stimulated with DNP-HSA (100 ng/mL) for 30 min and RBL-2H3 cells for 4 h. For the signaling analysis, the cells were pre-treated with various signaling inhibitors for 1 h and then stimulated with DNP-HSA. After incubation, the conditioned media and cell pellets were collected separately.

#### 2.12.1. β-Hexosaminidase Assay

The cell lysate was collected by lysing the cell pellets with 0.5% Triton X-100. The cell lysate and conditioned medium were incubated with substrate buffer (containing 0.1 M sodium citrate and 1 mM 4-nitrophenyl-N-acetyl-β-d-glucosamide, pH 4.5) at 37 °C for 1 h. After adding the stop solution (containing 0.1 M Na_2_CO_3_/NaHCO_3_), the absorbance was measured using microplate reader at 405 nm wavelength.

#### 2.12.2. Histamine Assay

As previously described, histamine levels in serum and conditioned media were measured [17]. First, the serum or conditioned media was reacted with 0.1 N HCl and 60% perchloric acid and centrifugated. Next, the supernatant was mixed with 5 M NaOH, 5 M NaCl, and *n*-butanol, vortexed, and centrifugated. After collecting the first fraction from the supernatant, it was mixed with 0.1 N HCl and *n*-heptane, vortexed again, and centrifugated. Finally, the second fraction collected from the bottom layer was incubated with *o*-phthaldialdehyde (Sigma-Aldrich) in an alkaline medium. The fluorescence intensity was measured at an excitation wavelength of 360 nm and an emission wavelength of 440 nm using a fluorescence spectrometer.

### 2.13. Quantitative Polymerase Chain Reaction (qPCR)

mBMMCs were sensitized with anti-DNP-IgE (50 ng/mL) for 16 h and stimulated with DNP-HSA (100 ng/mL) for 30 min. According to the manufacturer’s protocol, total cellular RNA was isolated (RNAiso Plus kit, Takarabio, Shiga, Japan), and *q*PCR was performed as previously described [17]. Used primer sequences are listed in Appendix A.

### 2.14. Statistical Analyses

All statistical data were analyzed using Prism 6 statistical software (GraphPad Software, Inc., San Diego, CA, USA). The results are expressed as the means ± standard error of the mean of three independent in vitro experiments and two independent in vivo experiments. Treatment effects were analyzed using a one-way analysis of variance followed by Sidak’s multiple comparisons test except for Figure 1A (*t*-test).

## 3. Results

### 3.1. Expression and Activation of IRF3 in IgE/Ag-Stimulated PCA Tissue and Mast Cells

At first, we analyzed the association of IRF3 in mast-cell-associated allergic disease using an anti-DNP-IgE/DNP-HSA-induced PCA mouse model. Histologically, anti-DNP-IgE/DNP-HSA-induced PCA tissues showed increased ear thickness and mast cell numbers (Figure 1A). The expressions of total IRF3 and phosphorylated IRF3 in the ear tissues were evaluated using Western blotting. IRF3 has phosphorylation sites on seven serine and threonine residues [18]. Among the sites, the Ser^386^ and Ser^396^ residues are critical for dimerization and nuclear translocation. In more detail, while the Ser^386^ residue is a complementary phosphorylation site for association with CBP/p300, the Ser^396^ residue is a more important site for activation [19]. Thus, we analyzed the phosphorylation of the Ser^396^ residue to verify IRF3 activation. PCA-induced tissues markedly increased the expression of the phosphorylated form as well as the total form of IRF3 (Figure 1B).

Typically, the stimulation of DNP-HSA in mast cells sensitized with anti-DNP-IgE and DNP-HSA induces activation of the mast cells. Therefore, we investigated whether the activation of IRF3 is involved in the mast cell activation process. The treatment of DNP-HSA in IgE-sensitized mBMMCs significantly induced the phosphorylation of IRF3 at even 1 min and was sustained for 5 min. Furthermore, the nuclear translocation (N-IRF3) for transcriptional activation increased until 15 min and decreased after (Figure 1C). Likewise, even though the exact time was different, RBL-2H3 also increased the phosphorylation and nuclear localization of IRF3 by DNP-HSA treatment (Figure 1D).

In turn, we analyzed the spatial expression pattern of activated IRF3 in the degranulation process of mast cells. Connective tissue-type mast cells sensitized with IgE are activated and degranulated by antigen. Tryptase, a major component of granules, is an indicator molecule for detecting mast cell activity As RBL-2H3 cells show low or undetectable tryptase expression [20], only mBMMCs were used for the analysis. Based on the phosphorylation timing of IRF3, we performed IF staining in mBMMCs stimulated for 2 min and 5 min (Figure 1E). In unstimulated conditions, tryptase intensively showed in the cytoplasm, but phosphorylated IRF3 (p-S^396^-IRF3) was not detected. After DNP-HSA stimulation, tryptase was still expressed in the cytoplasm, but the fluorescence intensity decreased at 2 min. After 5 min, it scattered throughout the cells. These results imply the degranulation process of mast cells. This was supported by measuring histamine levels in the conditioned media. The histamine level was significantly increased at 5 min stimulation (Appendix A). In comparison, the p-S^396^-IRF3 expression was increased in the cytoplasm at 2 min and spread throughout the cell at 5 min. The expression showed a spatially co-localized pattern with tryptase during mast cell activation. These results suggest that the activation of IRF3, including phosphorylation and nuclear translocation, might be involved in IgE/antigen-stimulated allergic responses and mast cell activation.

### 3.2. The Activity of IRF3 Depending on FcεRI-Mediated Signaling Pathway

Connective tissue-type mast cells are activated after the crosslinking of IgE and FcεRI upon repetitive exposure to an allergen [21]. The antigen binding on the IgE-FcεRI complex activates the downstream molecules, such as Src family kinases Lyn and Fyn. These activated Src family kinases subsequently activate Syk and activate the following downstream signaling cascades, such as PLCγ/PKC and PI3K/Akt [6]. Thus, we analyzed the molecular position of IRF3 in the FcεRI-mediated signaling pathway using several signaling inhibitors: the Src family kinase inhibitor, PP2; PI3k inhibitor, LY 294002; PLCγ inhibitor, U 73122; and Akt inhibitor, Akt inhibitor IV. Preliminarily, we evaluated the cytotoxicity of these inhibitors in mBMMCs and RBL-2H3 cells. PP2, LY 294002, and U 73122 did not show cytotoxicity up to 10 µM in both cell types. However, the Akt inhibitor IV exhibited cytotoxicity at 5 and 10 µM (Appendix A). Therefore, we used 1 µM of Akt inhibitor IV and 5 µM of PP2, LY 294002, and U 73122 in further experiments. Next, we verified the inhibitory effect on degranulation by measuring histamine (Appendix A) and β-hexosaminidase (Appendix A) levels. The pre-treatment of PP2, LY 294002, U 73122, and Akt inhibitor IV inhibited mast cell degranulation in IgE/Ag-stimulated cells. β-Hexosaminidase was also used as a marker of mast cell degranulation [22]. Then, we analyzed the effect on the phosphorylation and nuclear localization of IRF3 by the pre-treatment of inhibitors in DNP-HSA-induced mast cell activation. All inhibitors effectively suppressed the expression of p-S^396^-IRF3 and N-IRF3 in mBMMCs (Figure 2A) and RBL-2H3 cells (Figure 2B). These findings indicate that IRF3 activity is regulated as a downstream molecule of the FcεRI-mediated signaling pathway during mast cell activation.

### 3.3. The Effect of IRF3 in the Production of Allergic Inflammatory Mediators in IgE/antigen-Activated Mast Cells

Mast cells secrete various inflammatory mediators, including histamine, β-hexosaminidase, and inflammatory cytokines, to induce allergic reactions after activation [4]. Therefore, we investigated the effect of the alteration of IRF3 expression in producing allergic inflammatory mediators using siRNA and plasmid DNA for IRF3. Preliminarily, we verified the expression level of IRF3 in transfected cells with siRNA and WT plasmid DNA for IRF3 using Western blotting (Appendix A). The knockdown of IRF3 significantly reduced the levels of histamine and β-hexosaminidase in the conditioned media. In addition, TNF-α and IL-6, the inflammatory cytokines enhancing allergic inflammation, were also reduced (Figure 3A). In contrast, the overexpression of IRF3 increased the histamine, β-hexosaminidase, TNF-α, and IL-6 levels compared with the vector-transfected cells (Figure 3B).

Next, we further evaluated changes in the production of allergic mediators and inflammatory cytokines by DNP-HSA using WT and *Irf3* KO-derived mast cells. mBMMCs were prepared from WT and *Irf3* KO mice as described in the Materials and Methods section. Both mBMMCs showed no significant difference in the expression of the essential surface markers, FcεRI and c-kit (CD117) (Figure 4A). Simultaneously, we verified *Irf3* KO by analyzing the complete loss of IRF3 protein and gene expression (Figure 4B). *Irf3* KO mBMMCs by DNP-HSA stimulation showed reduced histamine and β-hexosaminidase, TNF-α, and IL-6 levels in the conditioned media (Figure 4C,D) compared with the WT. In addition, the gene expression of inflammatory cytokines, including TNF-α, IL-6, and IL-4, was dramatically reduced in the KO mice (Figure 4E). These data suggest that IRF3 regulates the production of secretory mediators as well as the gene expression of inflammatory cytokines.

### 3.4. Irf3-Deficiency Alleviates Mast-Cell-Mediated Anaphylactic Responses

We further investigated the effect of IRF3 deficiency in anaphylaxis models. During allergic reactions, the FcεRI-mediated activation of mast cells contributes to the pathophysiology of anaphylaxis by secreting histamine and other inflammatory mediators [23,24]. Therefore, we analyzed changes in the PCA reaction using WT and *Irf3* KO mice. PCA reactions are characterized by diffuse amounts of Evans blue pigmentation in local tissues from increased vascular permeability and ear swelling after antigen challenge [25]. Compared with the WT mice, *Irf3* KO mice showed reduced ear swelling and absorbance of Evans blue dye (Figure 5A). Moreover, mast cell infiltration was also decreased in *Irf3* KO mice (Figure 5B). In the OVA-induced ASA, repeated exposure to OVA induces allergic reactions that are enhanced by the production of OVA-specific IgE, IL-4, and histamine [23,24]. *Irf3* KO mice exhibited reduced hypothermia, a representative symptom of ASA (Figure 5C). In addition, OVA-challenged *Irf3* KO mice showed markedly reduced serum histamine, IL-4, total IgE, and OVA-specific IgE levels compared to the WT (Figure 5D–G). These results suggest that IRF3 might be a novel regulator to control allergic inflammation.

### 3.5. IRF3 Activity Promotes the Post-Translational Processing of HDC during Mast Cell Activation

The change in the secretion of histamine, a non-protein granule content, by IRF3 indicates that IRF3 possesses a molecular function in addition to its transcriptional activation role. Therefore, we investigated how IRF3 could regulate mast-cell-mediated allergic responses and the secreted level of granule contents from mast cells. HDC is an enzyme that catalyzes the synthesis of histamine from histidine. The 74-kDa HDC is located in the cytosol, translated to an active form of 53-kDa, and localized to the granule fraction [8,26]. The 74-kDa HDC is dominant in macrophages, whereas the 53-kDa HDC is detected in granule-containing cells, including RBL-2H3 cells [27]. *Hdc*-deficient mice failed to synthesize histamine and exhibited reduced or absent IgE-mediated anaphylactic reactions as well as severely decreased granule content [8,28]. Therefore, we investigated whether the alteration of IRF3 affects the activation of HDC. PCA-induced WT ear tissues increased the expression of cleaved HDC, whereas *Irf3* KO mice did not show the cleaved form (Figure 6A). To validate this finding in vitro, we analyzed HDC expression in DNP-HSA-stimulated mBMMCs and RBL-2H3 cells. Previous studies demonstrated that the cleaved forms of HDC protein are detected in 46-, 53-, 55-, and 63-kDa [8,29]. The intermediate forms, including 53-, 55-, and 63-kDa, have enzymatic activities but not 46-kDa. Therefore, the analysis ruled out changes in the expression of 46-kDa forms. Based on this report, we clearly detected a 63-kDa form of HDC in mBMMCs and a 55- and 53-kDa form in RBL-2H3 cells of DNP-HSA stimulation (Figure 6B). This cleaved form of HDC was suppressed in IRF3 knockdown cells (Figure 6C) by stimulation for 7 min, whereas it was increased in wtIRF3-overexpressing cells (Figure 6D). Interestingly, in relative band intensity analysis, we discovered the alteration of IRF3 significantly affected the cleaved form of HDC. These results suggest that IRF3 is an essential upstream molecule for the post-translational processing of HDC and, as a result, could regulate the level of granule components secreted from mast cells through HDC processing.

## 4. Discussion

In this study, we newly demonstrated the functional importance of IRF3 in mast-cell-mediated allergic disease and revealed the action mechanism of IRF3 in the granule content production by IgE/Ag-induced mast cell activation. Mast cells are primary cells in many immunological diseases and play a vital role in IgE-mediated allergic responses resulting in life-threatening anaphylaxis [30]. Repeated exposure to allergens results in the activation of the FcεRI-mediated signaling pathway and secretion of pre-formed secretory allergic mediators, including histamine and proteases, via degranulation [31]. The released allergic mediators induce pathophysiological responses in local and systemic tissues by increasing the permeability of the epi- and endothelium and causing circulatory changes [32]. For these reasons, FcεRI-mediated signaling pathways and mast cell-related allergic inflammatory mediators have been proposed as putative targets for reducing the risk of allergic inflammation [4,31].

Several studies have focused on the transcription factors controlling allergic inflammation by targeting mast cells [8,9,14,33,34]. Transcription factors are expressed in various immune cells, and they regulate the inflammatory response by modulating the expression of inflammatory cytokines [35]. IRF3 is a transcription factor that mediates the production of numerous inflammatory cytokines and plays an important role in the pathogenesis of several diseases [11,12,14]. Although the IRF family comprises IRF1–IRF9, several comparative studies have implicated IRF3 in allergen-associated inflammation [11,13,14]. In experimental models, *Irf3*-deficient mice could not develop allergic airway inflammation, whereas *Irf7*- and *Irf4*-deficient mice developed allergic responses and Th2 cytokine expression similar to WT mice [14,15]. In addition, *Irf3*-deficient mice could not produce IgE during OVA-induced adaptive-immune responses [15]. The evolved implication of IRF3 in the production of allergen-specific-IgE and Th2-mediated allergic inflammation has provided a foundation for further investigation into mast-cell-mediated allergic inflammation.

The transcriptional and non-transcriptional activity of IRF3 in other inflammatory disease models [11,36] has enabled us to investigate its role in mast-cell-mediated allergic inflammation. In general, IRF3 commutes between the cytoplasm and nucleus. Following activation, the cytoplasm-residing IRF3 is phosphorylated and subsequently translocated to the nucleus, leading to gene transcription [37,38]. Consistent with these findings, we found that the activation of p-S^396^-IRF3 and N-IRF3 occurs in mast cells after immunological stimulation via the crosslinking of IgE and FcεRI. Here, we extended these findings to show that p-S^396^-IRF3 co-localized with the retained granules (tryptase) in the cytoplasm 2 min after Ag stimulation, emphasizing the importance of activated IRF3 in mast cells. These results provide insight into the potential role of IRF3 in mast cell function and add reasonable information for approaches toward potential therapeutic intervention. A previous study showed that the PLCγ2-IP3-calcium signaling cascade activates IRF3 in lipopolysaccharide-induced macrophages [39]. Our study also demonstrated that the FcεRI-mediated signaling pathway induces phosphorylation and nuclear localization of IRF3 via the Syk-PLCγ and PI3k-Akt pathways. The FcεRI-mediated downstream signaling proteins contribute to the production of allergic inflammatory mediators [40].

Secreting allergic inflammatory mediators from mast cells in response to immunologic stimulation is instrumental in promoting allergic inflammation [40]. Based on this notion, we demonstrated that the knockdown or overexpression of IRF3 in mast cells altered the production of allergic inflammatory mediators in stimulated mast cells. These findings suggest that IRF3 activity in mast cells regulates inflammatory mediator production. Furthermore, these findings suggest the role of both the non-transcriptional and transcriptional activity of IRF3 in mast cells.

Previous studies indicated that IRF3 is required for allergic responses [14,15]. In agreement with previous findings, *Irf3* KO mice showed impaired local and systemic anaphylactic reactions compared with WT mice. Moreover, the results of an in vitro study using Ag-stimulated mBMMCs were consistent with the phenotypic characteristics of the in vivo models, indicating the involvement of IRF3 in the production of mast-cell-mediated inflammatory mediators. However, the *Irf3* KO mice used in our study are accompanying *Bcl2l12* KO. Therefore, to rule out the effect of *Bcl2l12* KO in the results using *Irf3* KO mice, we verified the effect of BCL2L12 on mast cell activity in vitro. WT mBMMCs were transfected with siRNA for *Bcl2l12* or *Irf3*. The knockdown level of Bcl2l12 was assessed using qPCR (Appendix A). While *Irf3* knockdown cells showed significantly reduced histamine, β-hexosaminidase, and inflammatory cytokines production, no significant effects were observed in *Bcl2l12* knockdown cells (Appendix A). Based on these results, we concluded that *Bcl2l12* had not affected the inhibitory responses in the study using *Irf3* KO mice.

Histamine released from mast cells is the hallmark of allergic inflammation and enhances the secretion of other secretory granule contents, including β-hexosaminidase and serotonin [41,42]. It is synthesized from histidine by the enzymatic activity of active HDC [8]. Previously, HDC-deficient mast cells showed reduced granule phenotypes and decreased histamine and β-hexosaminidase levels [41,43]. These studies imply that suppressing the post-translational processing of HDC might suppress histamine synthesis in mast cells [26,29]. In our study, the modulation of IRF3 expression induces changes in the production of histamine and β-hexosaminidase and cleavage of HDC possessing enzymatic activity. As is well known, the primary function of HDC enzyme activity is the synthesis of histamine. Therefore, HDC cleavage by IRF3 activity must have affected histamine secretion. Moreover, a recent study suggested that HDC might be involved in granule maturation [28]. However, they failed to explain the detailed mechanism of how HDC regulates granule maturation in the study. Based on previous reports and extended by our findings, we suggest that IRF3 could the post-translational processing of HDC, thereby regulating the production of granule contents in mast cells and the IgE-mediated allergic responses. As this study focuses on the novel role of IRF3 activity in mast cell activation, further studies on the control of the degranulation mechanism by HDC activity are needed.

Taken together, this study demonstrated that IRF3 activity is regulated as a downstream molecule of the IgE/FcεRI signaling cascade. It is verified that the alteration of IRF3 expression regulates HDC processing, the release of granule contents, such as histamine, β-hexosaminidase, and cytokines, and consequently, an allergic reaction. Based on these results, we can propose that Ag-stimulation leads to the activation of IRF3 through the IgE/FcεRI signaling pathway. The activated IRF3 regulates HDC processing in the cytoplasm and acts as a transcription factor after nuclear translocation.

## 5. Conclusions

In conclusion, our results suggest that activated IRF3 mediates FcεRI-involved signaling upon mast cell activation and promotes allergic inflammation by accompanying the activation of HDCs. The activity exerted by IRF3 on mast cell activation and cytokine production may have significant effects on IgE-FcεRI-mediated activated mast cells that are produced in patients with allergic inflammation. New approaches, such as the ex vivo expression of IRF3 in human allergic conditions, will be required to elucidate the precise role of IRF3 and test these findings in pre-clinical settings. However, our study could provide a new target to cure mast-cell-mediated allergic diseases, including asthma, allergic rhinitis, and sinusitis.

## Figures and Tables

**Figure 1 cells-12-01493-f001:**
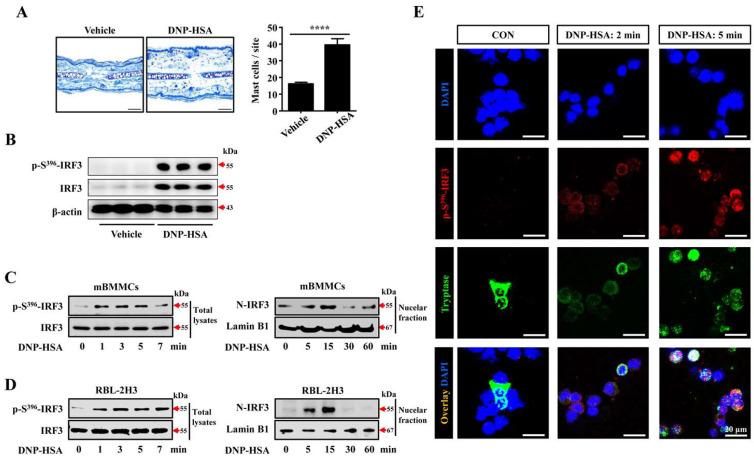
Expression of IRF3 in PCA-induced tissue and activated mast cells. Mice (*n* = 3/group) were sensitized with IgE and challenged with DNP-HSA to induce a PCA reaction. (**A**) Images of Toluidine blue-stained tissue obtained in ×200 magnification. Scale bar, 50 μm. The number of mast cells counted in five random sites. (**B**) The expression of p-S^396^-IRF3, IRF3, and β-actin in PCA-induced ear tissues. Anti-DNP-IgE-sensitized mBMMCs and RBL-2H3 were treated with DNP-HSA for the indicated time. (**C**) The expression of p-S^396^-IRF3, IRF3, and lamin B1 in the total and nuclear fraction of (**C**) mBMMCs and (**D**) RBL-2H3. (**E**) Immunostaining images for p-S^396^-IRF3 and tryptase of mBMMCs stimulated with DNP-HSA for the indicated time. Images were visualized at ×400 magnification. Scale bar, 20 μm. Blue; DAPI, Red; Alexa 594, Green; Alexa 488. The graph represents the means ± standard error of the mean. **** Significant difference at *p* < 0.0001.

**Figure 2 cells-12-01493-f002:**
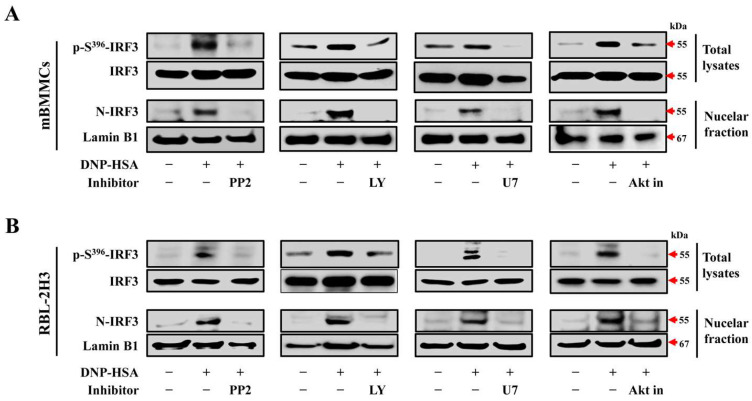
The effect of FcεRI-mediated signaling pathway in the activation of IRF3. Anti-DNP-IgE-sensitized mBMMCs and RBL-2H3 were pre-treated for 1 h with each inhibitor, including PP2 (5 µM), LY 294002 (LY, 5 µM), U 73122 (U7, 5 µM), and Akt inhibitor IV (Akt in, 1 µM), and then stimulated with DNP-HSA for 5 min for p-S^396^-IRF3 and 15 min for N-IRF3. The expression of p-S^396^-IRF3, IRF3, and lamin B1 in the total and nuclear fraction of (**A**) mBMMCs and (**B**) RBL-2H3.

**Figure 3 cells-12-01493-f003:**
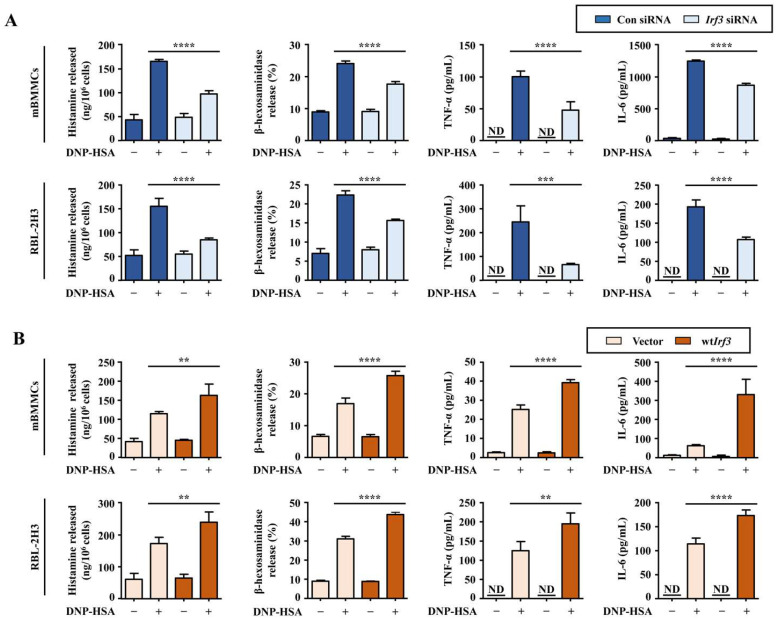
The effect of *Irf3* knockdown and overexpression in DNP-HSA-stimulated mast cells. siRNA or plasmid DNA-transfected cells were sensitized overnight with anti-DNP-IgE and stimulated with DNP-HSA for 30 min for mBMMCs and 4 h for RBL-2H3 cells. Histamine levels were measured using a fluorometric histamine assay. β-hexosaminidase level was measured using β-hexosaminidase substrate buffer. ELISA was used to determine the levels of TNF-α and IL-6. For ELISA, *Irf3* knockdown mBMMCs were stimulated with DNP-HSA for 8 h, and *Irf3*-knockdown RBL-2H3 cells were stimulated with DNP-HSA for 6 h. (**A**) Level of inflammatory mediators in *Irf3*-knockdown mBMMCs and RBL-2H3 cells. (**B**) Levels of inflammatory mediators in *Irf3*-overexpressing mBMMCs and RBL-2H3. Each dataset presents as the means ± standard deviation of the mean (*n* = 3). Significant difference at *p* ** < 0.01, *p* *** < 0.001, and *p* **** < 0.0001. ND: not detected.

**Figure 4 cells-12-01493-f004:**
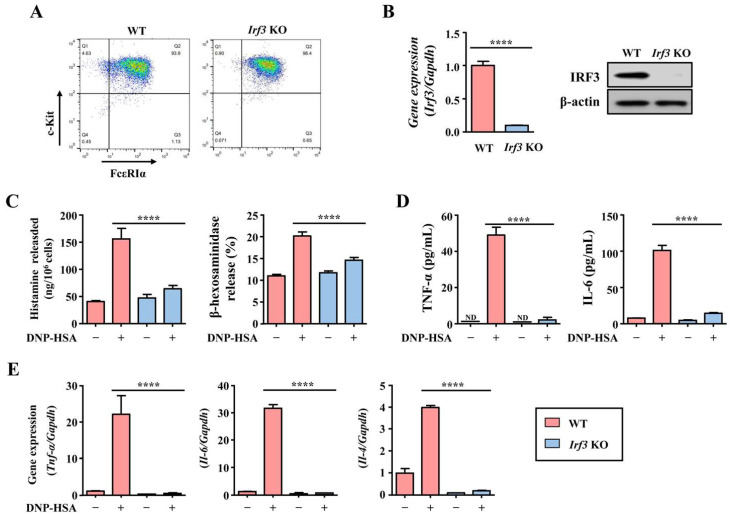
The effect of Irf3-deficiency in DNP-HSA-induced mast cell activation. Mouse bone marrow cells were isolated and differentiated into matured mast cells for 4 weeks as described in Materials and Methods. (**A**) Analysis of mast cell phenotypes using flow cytometric analysis. The color changes (blue>green>yellow< red) display the distribution intensity. (**B**) The gene and protein expression of IRF3 in primary cultured mBMMCs. WT and *Irf3* KO-derived mBMMCs were sensitized with anti-DNP-IgE and then stimulated with DNP-HSA. (**C**) Histamine and β-hexosaminidase levels at 30 min after stimulation. (**D**) The levels of TNF-α and IL-6 at 8 h after stimulation. (**E**) The gene expression of TNF-α and IL-6 at 30 min after stimulation. Each dataset presents as the means ± standard deviation of the mean (*n* = 3). **** Significant difference at *p* < 0.0001. ND: not detected.

**Figure 5 cells-12-01493-f005:**
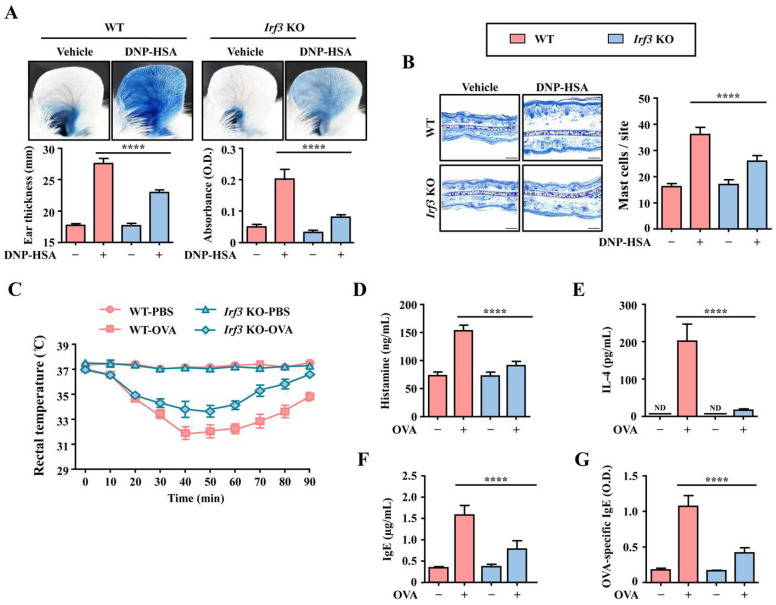
The effect of *Irf3*-deficiency in the anaphylaxis model. Passive cutaneous anaphylaxis reaction was induced in WT and *Irf3* KO mouse ear skin (*n* = 5/group) as described in Materials and Methods. (**A**) The representative photographic images of the ear, absorbance values of extracted dye from the ear, and ear thickness. (**B**) Images of Toluidine blue-stained tissue obtained in ×200 magnification. Scale bar, 50 μm. Active systemic anaphylaxis was induced in WT and Irf3 KO mice (*n* = 5/group) as described in Materials and Methods. (**C**) The changes in rectal temperature of OVA-challenged mice for 90 min. (**D**) Serum histamine levels. (**E**) Serum IL-4 levels. (**F**) Serum total IgE levels. (**G**) Serum OVA-specific IgE levels. Each dataset presents as the means ± standard error of the mean (*n* = 5). **** Significant difference at *p* < 0.0001. ND: not detected.

**Figure 6 cells-12-01493-f006:**
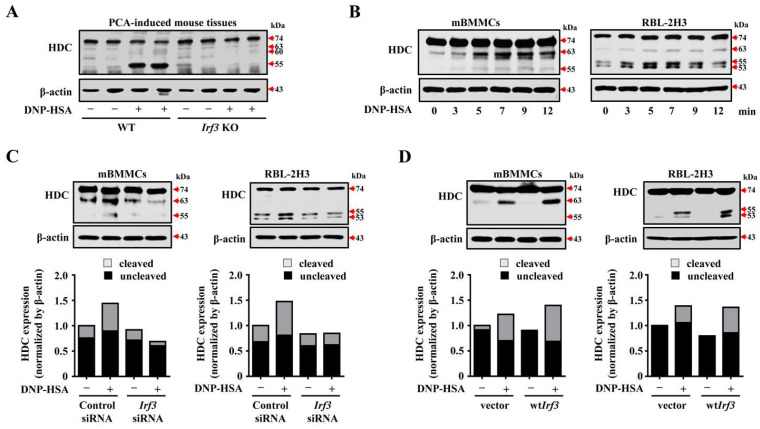
The effect of IRF3 in HDC post-translational processing. (**A**) The expression of HDC and β-actin in the passive cutaneous anaphylaxis-induced tissue. (**B**) Anti-DNP-IgE-sensitized mBMMCs (2 × 10^6^ cells/well in 6-well plates) and RBL-2H3 (1.5 × 10^6^ cells/well in 6-well plates) were stimulated with DNP-HSA. The expression of HDC and β-actin after the stimulation for the indicated time. (**C**) siRNA or (**D**) plasmid DNA-transfected mBMMCs and RBL-2H3 cells were sensitized with anti-DNP-IgE overnight and then stimulated with DNP-HSA. The expression of HDC and β-actin (**upper images**) and relative band intensities of HDC (**lower graph**).

## Data Availability

The datasets generated for this study are available on request from the corresponding author.

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
