# Peer review of "IRF3 Activation in Mast Cells Promotes FcεRI-Mediated Allergic Inflammation"

_cells, 2023, doi:10.3390/cells12111493_

Round 1

Reviewer 1 Report

The authors aimed to show a novel function of IRF3 as an important factor  inducing mast cell activation and as an upstream molecule for HDC activity“ independent of its role as a transcription factor.”

The data presented is straightforward; however, it raised some concerns that preclude the publication in this form:

Major points :

The authors claimed that the IRF3 actions are independent of its activity as a transcription factor; however,  figure 1 shows translocation to the nucleus. It needs clarification because the mechanism is not defined.

Authors show IRF3 affects the release of preformed mediators (Histamine, β-hexosaminidase) and gene expression IL6 and TNF. Which mechanism do the authors propose?

Figure 1. In figures 1 C and B, apart from lamin, the authors should show tubulin to show no cytoplasm contamination. Please, add molecular weights in all blots.

Is the N-IRF3 phosphorylated in the nucleus? How is translocation regulated (import and export to the nucleus)? Which kinase is involved in phosphorylation? Why is so transient the IRF3 expression in the nucleus?

Fig 1E. The authors mentioned: “The expression showed a spatially colocalized pattern with tryptase during mast cell activation.” Is  IRF3 in the granules and colocalizes with trypsin? However, at this time, is it also translocated to the nucleus? Is IRF3 released with trypsin? Please, clarify all these points.  

Figure 2. All inhibitors used inhibit IRF3 phosphorylation; although phosphorylation is in Ser, Src inhibition (affect tyrosine phosphorylation) is also affected. What pathway is proposed? Please explain and add the time of activation in this figure.

Figure 6. HDC degraded or cleaved are used indistinctly; however, the outcome may differ. This figure is confusing after reading the text in the body of the manuscript. Cleaved is the active form. Is then IRF3 affecting the activated form? If IRF3 is involved in the degraded form  (inactive form, as the authors mentioned), how does it explain the reduction of histamine after IRF3 knockdown? This issue needs clarification. Reference 29 mentions”, histamine synthesis is augmented through the post-translational cleavage of HDC, which caspase-9 mediates”. Do the authors check Caspase 9 (pro and active form) in the context of IRF3 (silencing and overexpression )?

Figure 6 needs statistical analysis.

It is hard to check the original blots because they are in separate files, not labeled, with no molecular weight added. Several bands show up that are not in the figure. Ex: Fig 1B pIRF3 does not seem the same. Can they put the original blots well-labeled for each figure together in a word to check them easily?

Minor points:

-There is no supplementary figure 1 (?). Renumber the supplementary for logical order.

-Explain the protocol for Histamine detection in more detail.

Reviewer 2 Report

In this paper, the authors examined the roles of IRF3 on mast cell activation and HDC processing. They found that IRF3 was involved in degranulation as well as cytokine production and processing of HDC. Thus IRF3 plays important roles via transcriptional and non-transcriptional activity.

Major points:

1) The effects of irf3 siRNA on degranulation and cytokine expression were clear, but that on processing of HDC was marginal.  What difference do the authors think between signal transduction pathway to degranulation and activation of processing.

2) In irf3 KO mice, were there any difference of granule formation and histamine content in mast cells?

3) The authors should explain about the molecular mechanisms of non-transcriptional activity of IRF3 on mast cell degranulation and processing of HDC.

Round 2

Reviewer 1 Report

The authors have address many of the aspects raised by reviewer.

However I still have some concerns:

I can not find the word with the original blots embebed in it, mentioned in point 9,  either the graphical abstract with the proposed mechanism.  

The authors claim that " Colocalization of IRF3 and tryptase implies that the activity of IRF3 is required for granules to mature and release". That sentence is a overstatement and should be corrected. The novel IRF3 mechanism apart from its activity in nucli unfortunately remains not clear. 

Author Response

We appreciate the suggestions made by the reviewer. All suggestions by the reviewer have been incorporated in the revised draft.

Following is a detailed list of all changes made, referring to the points raised by the reviewer.

Point 1: I can not find the word with the original blots embebed in it, mentioned in point 9, either the graphical abstract with the proposed mechanism.  

Response 1: We are very sorry for this trouble. The related files did not seem to be uploaded in the proper section (last time, it was uploaded as a supporting image to the main submission page).

The re-revised version has been uploaded as a PDF file combined with raw images and a graphical abstract to the section of Author's Reply to the Review Report (Reviewer 1).

Point 2:The authors claim that "Colocalization of IRF3 and tryptase implies that the activity of IRF3 is required for granules to mature and release". That sentence is a overstatement and should be corrected. The novel IRF3 mechanism apart from its activity in nucli unfortunately remains not clear. 

Response 2: Thank you for the critical comment. We assume that the reviewer pointed out the following sentence on page 6, lane 270~271; "These results imply that the activity of IRF3 is required for granules to mature and release in the IgE-mediated mast cell activation process." 

We agree with the reviewer's claim. The provided results are insufficient for that conclusion. In the re-revised version, we tone-downed the indicated sentence as follows (page 6, lane 270~272): "These results suggest that the activation of IRF3, including phosphorylation and nuclear translocation, might be involved in IgE/antigen-stimulated allergic responses and mast cell activation."

Reviewer 2 Report

I assessed this version is now acceptable.

Author Response

N/A